# Tensile Properties of Aircraft Coating Systems and Applied Strain Modeling

Attilio Arcari *, Rachel M. Anderson, Carlos M. Hangarter, Erick B. Iezzi [ID] and Steven A. Policastro *

Center for Corrosion Science and Engineering, U.S. Naval Research Laboratory, 4555 Overlook Avenue SW, Washington, DC 20375, USA
* Correspondence: attilio.arcari@nrl.navy.mil (A.A.); steven.a.policastro.civ@us.navy.mil (S.A.P.)

**Abstract:** In this work, we develop a structural model for the fracturing of an aircraft coating system applied to a complex airframe structure that includes aluminum panels and stainless-steel fasteners. The mechanical properties of the coating system, which consisted of an MIL-PRF-85582E, Type II, Class C1, two-part epoxy primer and an MIL-PRF-85285 Rev E, Type IV, Class H, two-part polyurethane topcoat, were measured before and after 8 months of atmospheric exposure. The loads applied to the coating occurred from local deformations of the fastener-panel system in response to flight stresses. Two types of flight stresses, compression dominated and tension dominated, were modeled. The degradation of the mechanical properties of the coating after atmospheric exposure increases the severity of cracking of the coating at a critical fastener–skin interface.

**Keywords:** finite element analysis; mechanical properties; fatigue; organic coating; failure mechanisms



## 1. Introduction

Aircraft skins generally consist of aluminum alloy sheets or extrusions made from UNS A9204 or UNS A97075 and fastened to other aluminum alloy structures, such as spars, ribs, or stiffeners. Countersunk fasteners, e.g., aluminum rivets, made from an alloy such as UNS A93003, or threaded stainless-steel screws, made from an alloy such as UNS S31600, have a head specifically designed to "sink in" the fastened part to be flush with the outer surface. They are typically used on surfaces exposed to airflow, e.g., to fasten wing skins to the underlying structure. Localized attacks at the fastener–skin interfaces has been reported as particularly important for aircraft skins [1]. These attacks occur when the galvanic couple between the dissimilar metals in the fastener–skin junction is exposed to electrolytes. Thus, corrosion has been found to occur on and around the countersunk surfaces of the aluminum skin [2,3].

Exfoliation corrosion is often observed as the first step in corrosion damage of the skin, progressing away from the sides of the fastener borehole parallel to the skin surface and eventually leading to fatigue crack initiation and propagation [4]. Exfoliation occurs as a first step because the aluminum skin has an inherent microstructural directionality due to the crystalline grain orientations formed during the forging process. Preventative measures, such as wet installation of fasteners in addition to specific surface preparation of the mating components, are employed. However, time and wear of the surfaces degrade this protection, leading to galvanic interactions and the start of corrosion damage.

Coating systems designed for aircraft structures are tasked with providing the principal means of protection against galvanic and uniform corrosion, but they perform multiple functions to do so [1]. They act as a barrier against direct contact with the atmospheric environment and inhibit corrosion when through-thickness coating damage occurs [5]. In addition, they must sustain corrosion protection in a wide range of environmental conditions, ranging from arid environments to sea salt spray, and are subjected to significant temperature fluctuations. Exposure temperatures range from sub-freezing ($-95\ ^\circ$C/$-64\ ^\circ$F) when aircraft

fly at high altitude to near-boiling (93 °C/200 °F in some cases), depending on the geographical location and speed. In addition, coating systems must adhere to multiple substrate materials and maintain resilience against deformations, including bridging gaps between surfaces. Adhesion between the coating and aluminum substrates is usually enhanced by the application of a pretreatment that oxidizes the aluminum to thicken the oxide and improve the mechanical bond with the cured coating.

When coatings fail to provide adequate protection against electrolyte intrusion, corrosion damage follows. Once coating damage is detected, repairs are labor-intensive and time-consuming. Typical options include stripping and re-application of the coating system to the structure. For small areas of metallic surfaces, paint is removed by hand using medium-grade abrasives. For larger areas, chemical paint removal is still the most widely used method [1], though laser ablation is gaining wider acceptance. Once the coating is removed, the metal area is reworked to eliminate all traces of corrosion. In the last step, a new coating system is applied.

Thus, improving our understanding of the mechanical behavior of aircraft coatings in response to environmental exposures and flight stresses is important to help those responsible for aircraft maintenance make decisions on coating lifetimes and repairs.

To that end, this work focuses on modeling an aircraft coating system's response to tensile and compressive flight loads in a high-stress region on an aircraft. Loads to the coating system arise from displacements due to the relative motion of the aircraft skin and fastener heads in response to flight stresses. One high-stress location on airframes is located at the wing root [6], where the wing joins the aircraft body. A representative geometry was chosen that consisted of two aluminum plates and countersunk stainless-steel fasteners. Such a configuration is likely to be found near the wing root where skin panels are fastened to an underlying structural element.

We developed a finite element model (FEM) to identify the high deformation areas in the coating near a fastener under a complex loading mode and quantify the magnitude of the applied strains. We then investigated the coating deformations at a fastener–skin joint using numerical methods on a representative structural configuration, and we quantified the elastic properties of coating system components to determine the onset of cracking.

The critical stress–strain conditions that lead to cracking and coating failure are quantified through experimental testing of the materials in the coating system. These stress-strain conditions may be changed by the environment, e.g., exposure to UV light and temperature fluctuations from flight operations, which can lead to cracking and coating failure at earlier intervals than the planned service life of the coating [7–10].

The manuscript is laid out in the following manner: first, the experimental methods for measuring the elastic properties of the polymer coatings before and after atmospheric exposure are described. The FEM results and the calculated strains at the fastener–skin interface are presented. The elastic properties of the polymers are used as inputs for the fracture model to inform the model and determine the critical fracture stress. A crack propagation model is then used to study the progression of a crack through the coating layers, starting from the primer and propagating through the topcoat using the elastic properties for the coating layers.

## 2. Materials and Methods

### 2.1. Sample Preparation

Samples of the primer, topcoat, and full coating system were applied to the outer surface of polyvinyl fluoride (PVF) films (DuPont, Holliston, MA, USA) to enable the testing of free films of the coatings in tension. For primer samples, the films were spray coated using high-volume, low-pressure (HVLP) spray equipment (3M, St. Paul, MN, USA)) with a two-part water-reducible epoxy (44GN008A; PPG Industries, Pittsburgh, PA, USA) which included a barium chromate-based corrosion inhibitor that complied with the coating performance specification MIL-PRF-85582E, Type II, Class C1. For topcoat samples, PVF films were coated using HVLP spray equipment with a two-part gray (FED-STD-595 #36375) polyurethane

topcoat (99GY003; PPG Industries, Pittsburgh, PA, USA) that complied with the coating performance specification MIL-PRF-85285, Type IV, Class H. For full coating system samples, the primer was first applied to the PVF film and allowed to cure for a minimum of 24 h before the topcoat was applied. All samples were cured for 24 h before dry film thickness (DFT) measurements were made to determine the sample thicknesses. The layer thicknesses were consistent and representative of typical aircraft coatings. The primer layers were approximately 25 μm thick and the topcoat layers were approximately 100 μm thick, while the full system samples, composed of a single primer and topcoat layer, were approximately 125 μm thick. All samples were then allowed to further cure under ambient conditions, i.e., 20–22 °C, 40–60% R.H., for a minimum of 14 d before exposure testing in accordance with the manufacturer's specifications.

### 2.2. Outdoor Exposure Testing

The coating samples were left on the release paper and mounted to moisture-resistant high-density polyethylene (HDPE) sheets (McMaster-Carr, Robbinsville, NJ, USA) using threaded nylon fasteners with the sample sandwiched between nylon washers (McMaster-Carr, Robbinsville, NJ, USA). The HDPE sheets with samples attached were mounted so that the samples were inclined at a 45° angle from vertical on the atmospheric exposure racks at Fleming Key, FL. They were continuously exposed to atmospheric conditions for 5736 h, or approximately 8 months. The 45° angle exposure rack was designed to maximize irradiation by UV light while allowing rainwater to run off easily. An image of the exposure rack, with the coating samples mounted, is shown in Figure 1. The image includes the primer, topcoat, and full coating system samples. The two single layers of primer are shown in the bottom right corner of the exposure panel. Unfortunately, none of the primer samples survived the exposure period. The epoxies used for aircraft coating systems are brittle once cured and are susceptible to UV damage. The combination of weathering and UV degradation resulted in multiple brittle fractures, and only small pieces of the primer samples were found attached to the nylon fasteners. At the completion of the exposure period, the full coating system and the topcoat samples were shipped back for the machining of tensile samples. The topcoat and full coating system were cut into rectangular tensile samples with the same geometry used for the unexposed materials. Figure 2 shows the average environmental conditions, in 30 min increments, over the duration of the coating system testing at the atmospheric exposure site.

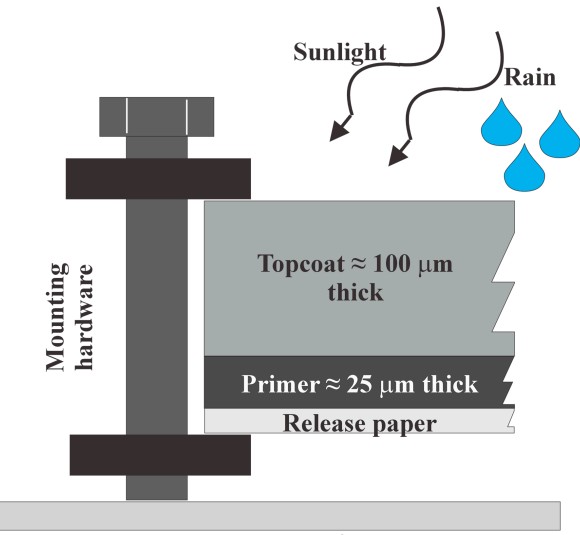

**Figure 1.** Setup for the atmospheric exposure testing of the primer, topcoat, and full coating system samples. This diagram shows the full stack-up of the topcoat and primer. The individual layers were applied directly to the release paper. The mounting plate was oriented at 45 ° to increase sunlight exposure and to allow water to run off the samples.

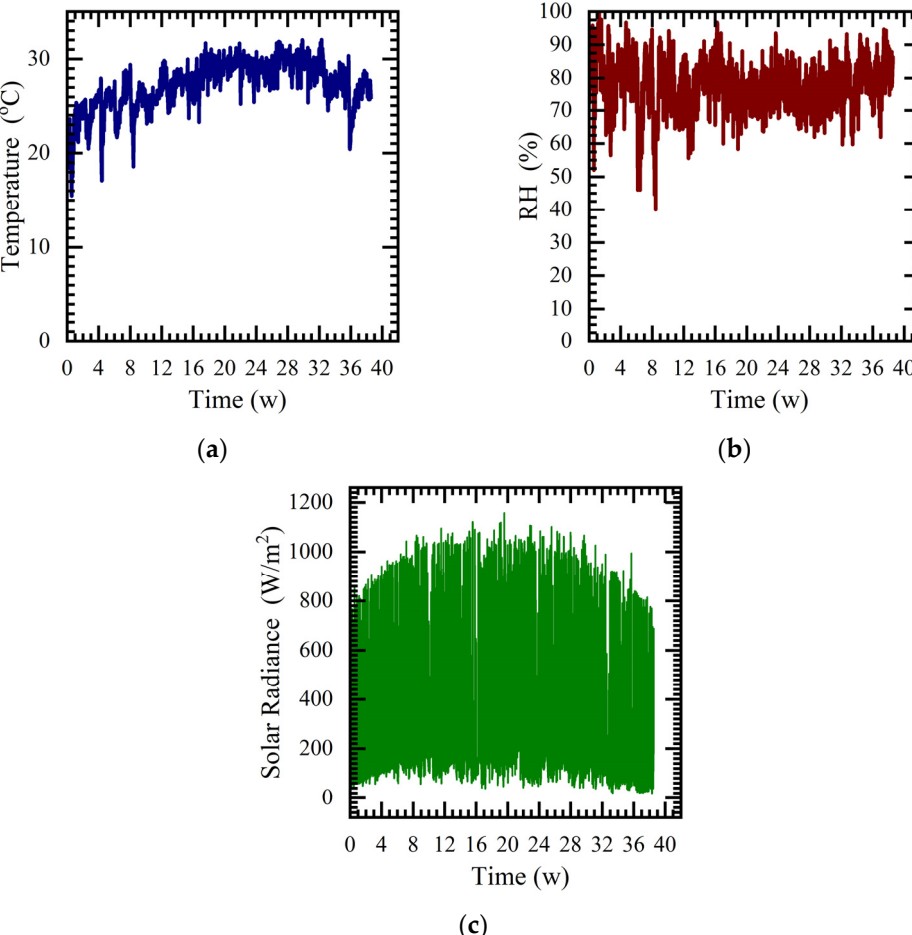

**Figure 2.** Environmental conditions at the atmospheric test site during the coating system exposures (time is expressed in weeks). (**a**) Temperature, (**b**) relative humidity, and (**c**) solar radiance.

### 2.3. Tensile Testing of Primer and Topcoat Materials

The PVF films were cut into 2.54 cm-wide samples to allow the coating samples to be detached and positioned in the testing machine. A Testometric load frame (Testometric, Rochdale, U.K.) with a 250 kgf load cell was set up to allow the gripping of 2.54 cm wide strips of the materials. Both the applied loads and displacements were measured during the test using a load cell and the crosshead displacement of the load frame [11]. The elastic properties for the various coating samples were determined before and after atmospheric exposure. The tensile testing setup is shown in Figure 3.

It was very difficult to avoid breaking the epoxy samples while handling them for mounting in the load frame and applying the grips. Observations from the data that were collected indicated an elastic behavior followed by brittle fracturing of the sample. The results for the testing of the topcoat were consistent and showed an initial linear behavior followed by a continuous decrease in the instantaneous modulus until final failure near 4% strain. The complete coating system demonstrated an initial elastic behavior, with a similar modulus as that measured for the topcoat alone, up to a point where a maximum load was measured, and the deformations started to increase significantly. Three tests of the completed coating system were performed. Failure occurred at around 4% strain for two of the three tests, but at around 2.5% for the third and final test.

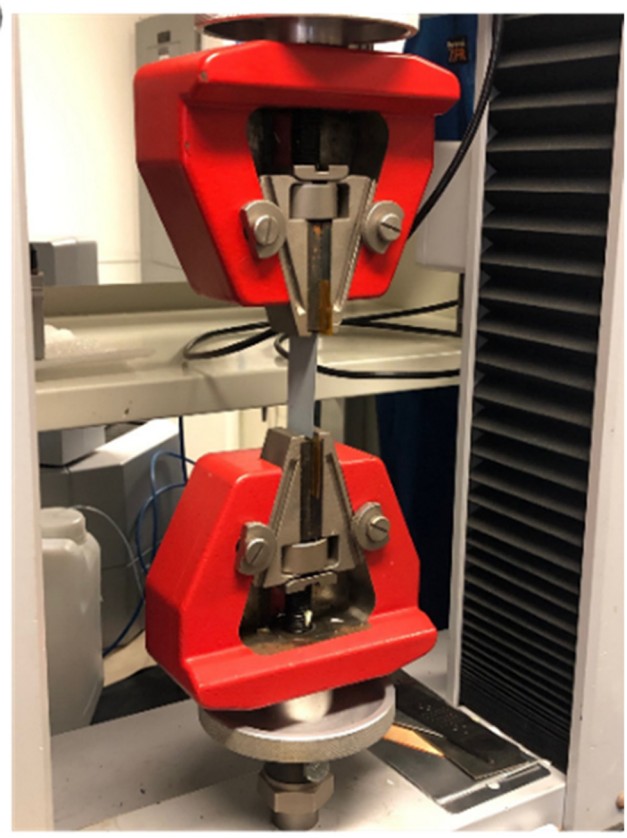

**Figure 3.** Test configuration for measuring the elastic properties of the different coating samples.

### 2.4. Coating System Stress Field Model

To simulate the local fastener-plate interface displacements in response to flight loads, a model geometry was constructed. The model configuration, shown in Figure 4, consisted of two aluminum plates, each with a 70 GPa elastic modulus, and 20 countersunk fasteners. The fastener material was stainless steel with an elastic modulus of 210 GPa. The fastener and fastener holes were countersunk with an angle of 100 degrees and a countersunk head height of 1.73 mm. The fastener shank and nut were modeled as a single part. The bolt was simplified as an integral part of the fastener to allow application of the desired amount of force as the preload. The coating layer was applied to the aluminum plate at the locations shown in Figure 4. The coating layer used a higher mesh density around fastener #6 to allow for a more refined characterization of the local deformations.

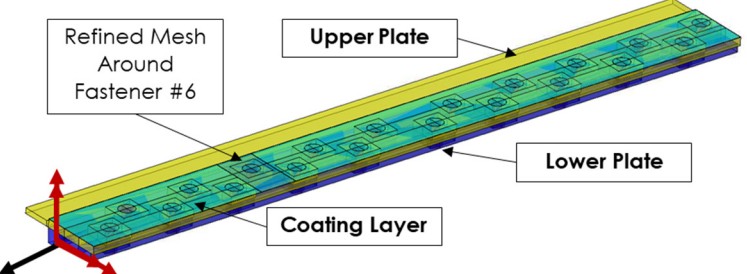

**Figure 4.** Geometry and components of the finite element model in this work.

The coating system was modeled as a bilayer structure consisting of a primer and topcoat that were 25 μm and 100 μm thick, respectively. A perfect bonding between the two layers and a perfect bonding of the primer with the aluminum plate and fastener is assumed. Note that these model assumptions closely mimic in-service conditions. Aluminum alloy surfaces receive a pretreatment or conversion coating that anodizes the aluminum oxide

layer to thicken it and to aid in the adhesion between the surface and the primer. The surface pretreatment was not explicitly included as a separate layer in the model. In addition, coating manufacturers specify a maximum delay for overcoating primers, after which topcoats cannot be applied. This is to ensure that there is good adhesion between the topcoat and primer.

The analysis was divided into two steps: the first step applied the preload to the fasteners and the second step applied the external loads to the entire model. Two sets of external loads were applied to the model: one which caused the upper plate to be under tension and one which caused the upper plate to be under compression. These types of loads correspond to events such as an aircraft landing or flying through wake turbulence, respectively. The Von Mises stress of the top aluminum plate away from the fasteners obtained for the tension-dominated case was approximately 1/3 of the yield strength, which is assumed to be 450 MPa for this alloy. For simplicity, we refer to "tension-dominated" and "compression-dominated" loads for the rest of the manuscript. These refer to the condition of the upper plate and define the upper and lower bounds of the deformations obtained, respectively. The loads are identical in magnitude for the two cases but reversed in sign. The applied loads consisted of a combination of two bending moments and an axial force to represent a typical mixed-mode condition of a structural member.

The installation of the fasteners and the application of preloads during the assembly process preceded the coating of the skin panels. This was simulated in the model by deactivating the coating layers during the preload. This deactivation step allowed the coating layer to conform to the final shape of the structure without causing any strains or stresses in the coating. This initial "inactive" state was included to ensure that the strains and stresses in the coating were only caused by the external loads. The net result was that the coating layer adhered perfectly to the contours of the fastened structure around the fastener heads and skin.

The areas between the fastener head and countersunk hole surface, between the fastener shank and hole surface, and between the nut and bottom plate surfaces were in contact, as shown in Figure 5. The contact properties were defined in the model as non-penetration in the perpendicular direction to the contact surface and friction in the tangential direction. The latter used a static coefficient of friction of 0.61 for clean and dry steel [12]. The areas between the upper and lower plate were also modeled with contact, defined as non-penetration in the perpendicular direction to the contact surface and friction in the tangential direction. The static coefficient of friction was 1.05 [12], representative of clean and dry aluminum surfaces.

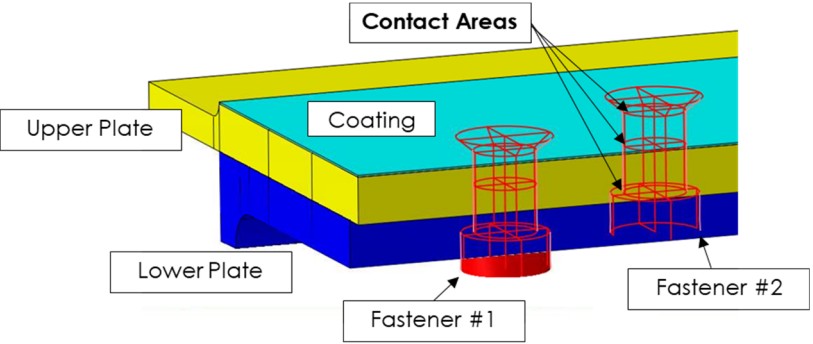

**Figure 5.** Detail of the model in this work with defined contact surfaces between fasteners, top, and bottom plates.

A sub-modeling strategy was used to refine the results around a single fastener: the displacements resulting from the application of the loads were used as boundary conditions for a sub-part of the model, specifically the coating layer on top of fastener #6. This enabled significant mesh refinement near the region of interest without paying a large penalty in

terms of computational time. The analysis was performed using Abaqus and CAE on an HPE SGI 8600 high-performance computing cluster.

We used the properties obtained from the tensile testing of the different coating layers to study the expected strains in the primer and topcoat layers of the coating system. The substrate materials were assumed to be able to withstand the applied deformations without failing. The next section includes a failure criterion to model the occurrence of cracking during the studied loading cases, given the observed fracture stresses and strains from tensile testing.

The primer layer was modeled as a linear elastic material with an elastic modulus obtained through analysis of the tensile tests. The topcoat was modeled as an elastic–plastic material with its initial modulus and yield stress obtained from analysis of the tensile tests. The plastic behavior was modeled with kinematic hardening. That is, the models used calculated stress–strain data as a reference for an elastic–plastic regression to find the points needed by the kinematic hardening model. The plasticity model used for modeling the topcoat material was based on the Von Mises yield criterion, given in Equation (1),

$$f(\boldsymbol{\sigma} - \boldsymbol{\alpha}) = \sqrt{\frac{3}{2}\left(\boldsymbol{S} - \boldsymbol{\alpha}^{dev}\right):\left(\boldsymbol{S} - \boldsymbol{\alpha}^{dev}\right)} = \sigma_y, \tag{1}$$

where $\sigma$ is Cauchy stress and $S$ is the stress tensor in the deviatoric space. $\boldsymbol{\alpha}$ and $\boldsymbol{\alpha}^{dev}$ are the back stress and the back stress in the deviatoric space, respectively, and $\sigma_y$ is the yield strength. The plastic flow always follows a direction perpendicular to the yield surface, as shown in Equation (2),

$$\dot{\boldsymbol{\varepsilon}}^{pl} = \frac{\partial f(\boldsymbol{\sigma} - \boldsymbol{\alpha})}{\partial \boldsymbol{\sigma}} \dot{\bar{\varepsilon}}^{pl}, \tag{2}$$

where $\dot{\boldsymbol{\varepsilon}}^{pl}$ is the plastic strain rate and $\dot{\bar{\varepsilon}}^{pl}$ is the equivalent plastic strain rate defined as $\dot{\bar{\varepsilon}}^{pl} = \sqrt{\frac{2}{3}\dot{\boldsymbol{\varepsilon}}^{pl}:\dot{\boldsymbol{\varepsilon}}^{pl}}$ [13]. The linear kinematic hardening model defines the back stress rate of change in Equation (3) as,

$$\dot{\boldsymbol{\alpha}} = C\dot{\bar{\varepsilon}}^{pl}\frac{1}{\sigma_y}(\boldsymbol{\sigma} - \boldsymbol{\alpha}) + \frac{1}{C}\boldsymbol{\alpha}\dot{C}, \tag{3}$$

where $C$ is a hardening parameter that in its simplest form describes the slope of the stress-plastic strain response of the material for a uniaxial tensile test [14].

### 2.5. Coating Cracking Model

A model was developed to include a fracture criterion for both the primer and topcoat layers based on the experimental fracture stress obtained in tensile testing. For the primer layer, failure occurred when the maximum principal stress calculated in the analysis was higher than a value of 32.0 MPa. For the topcoat layer, failure occurred when the principal stress was higher than 18.0 MPa. An enrichment function that allowed the elements to break upon satisfying the failure criterion was included by making use of the extended finite element method (XFEM) implementation in Abaqus. The use of XFEM allowed us to simulate the initiation and propagation of a crack.

## 3. Results

### 3.1. Tensile Testing of Unexposed Coatings

Tensile test results for the unexposed topcoat and full coating system are shown in Figure 6.

The elastic moduli for the topcoat and the full coating system were determined through linear interpolation of the stress and strains based on a 5.08 cm gauge length and with thicknesses as indicated above. The values are reported in Table 1, along with the estimated value of the elastic modulus for the primer material.

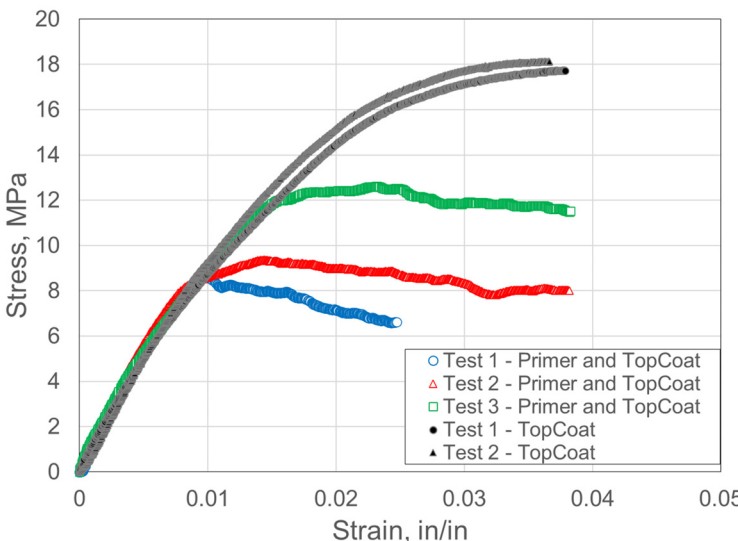

**Figure 6.** Tensile test results for the unexposed topcoat and full coating system samples. Test results for the unexposed primer samples are not included.

**Table 1.** Average values from tensile test results for primer and topcoat materials and for the full coating system.

| Coating Layer | Elastic Modulus (MPa) | Fracture Strain (%) |
|---|---|---|
| Full coating system | $1028.6 \pm 133.9$ | $3.5 \pm 1.1$ |
| Topcoat | $846.0 \pm 74$ | $3.8 \pm 0.2$ |
| Primer (estimated from Equation (1)) | 3102.7 | -- |

The elastic modulus of the primer layer was estimated through interpolation of the elastic modulus of the coating system and a modulus for the topcoat layer as calculated from the tensile tests of the topcoat free film. The resulting force for the sample with primer and topcoat is assumed to be given by the expression in Equation (4),

$$F_c = \varepsilon E_1 A_1 + \varepsilon E_2 A_2, \tag{4}$$

where $F_c$ is the force applied to the coating system and the applied displacements are $\delta_1 = \delta_2 = \delta_c$ and $\varepsilon = \delta_c / L$, with $L = 5.08$ cm. This implies that both the primer and topcoat are subjected to equal displacement during the test and that they are perfectly bonded at their interface. The modulus obtained through calculations for the primer is in line with values reported in the literature for epoxies [15].

It is interesting to note that the full coating system shows elastic–plastic characteristics as a combination of the higher modulus of the primer and the higher toughness of the topcoat. The modulus of the coating system is higher than the modulus of the topcoat alone and the failure strain is significantly higher than the failure strain of the primer, while still within the range of the failure strain for the topcoat.

### 3.2. Tensile Testing of Aged Coatings

The results of the tensile tests on the exposed, aged coatings, in comparison to the unexposed full stacks and topcoat, are shown in Figure 7. The test results for the full coating system were inconclusive. While the initial elastic modulus was generally consistent, the behavior at strains higher than 0.005 differed significantly among the three exposed samples. In addition, they showed a wide range of maximum stresses, ultimate tensile strains, and strains at failure.

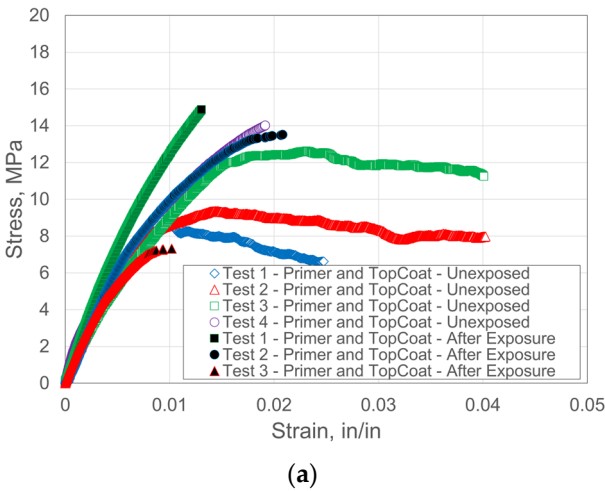 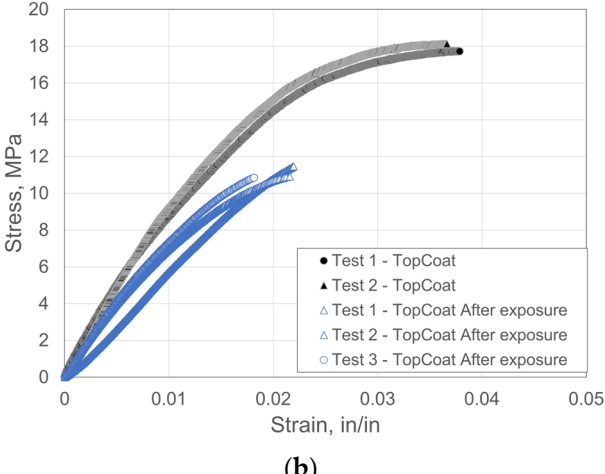

(**a**)                                   (**b**)

**Figure 7.** Tensile testing of (**a**) the full system of primer and topcoat coating after exposure to the atmospheric environment for 5736 h and comparison with unexposed tensile testing results, and tensile testing of (**b**) topcoat material coating after exposure to the atmospheric environment for 5736 h and comparison with unexposed tensile testing results.

In contrast, the tensile curves for the topcoats from each condition were consistent with each other, as shown in Figure 7b. The topcoat material showed a significant decrease in the tensile properties after exposure, with both a reduction in the elastic modulus and a significant reduction in the ultimate strain. The modulus averaged 722 MPa (SD = 36), a reduction of 14.7%, while the strain at failure averaged 2.04% (SD = 0.2), a reduction of 46.3% with respect to the unexposed conditions.

### 3.3. Model Results

A summary of the input for both materials with material properties obtained from the tensile behavior from Figure 6 is shown in Table 2.

**Table 2.** Elastic input for the constitutive relation of primer and topcoat materials in the model developed.

| Coating Layer | Elastic Modulus (MPa) | Poisson's Ratio |
|---|---|---|
| Primer | 3102.0 | 0.37 |
| Topcoat | 846.0 | 0.4 |

The stress, $\sigma$, and plastic strain, $\varepsilon_\mathrm{p}$, which are inputs for the elastic–plastic behavior of the topcoat using the kinematic hardening model discussed earlier are shown in Table 3.

**Table 3.** Elastic–plastic input for the kinematic hardening constitutive relation of the topcoat materials in the model developed.

| Stress (MPa) | Plastic Strain (mm/mm) |
|---|---|
| 13.79 | 0 |
| 15.169 | 0.002 |
| 16.548 | 0.0055 |
| 17.2375 | 0.0096 |
| 17.927 | 0.0138 |

A variety of interference fits and preload conditions were modeled. The interference fits were identified by the numbers 1–3, which corresponded to 0.05%, 0.15%, and no interference, respectively. Each interference fit could have one of three preloads, identified as a, b, or c. These preload conditions consisted of 50% preload, 75% preload, and 100%

preload, respectively. Thus, for example, Model 3c would identify a simulation with no interference fit and 100% preload applied to the fastener-skin joint.

The model number corresponds to the level of interference fit between the fastener size and hole diameter: (1) medium (0.05%), (2) high (0.15%), and (3) no interference. The different preload conditions are low (50%), medium (75%), and high (100%).

Our results show that high strains are experienced at the location where the relative displacement between the fastener head and the aluminum plate is highest. A larger relative displacement is obtained when tension-dominated loads were simulated, while a lower, but still significant, displacement was predicted for the case of compression-dominated loads.

The main deformations that determine the strains in the coating layer for the two loading cases studied are related to the relative motion of the fastener head and countersunk surface of the plate. This motion is shown in Figure 8 for the case of a 75% preload and the highest interference fit modeled. The displacements are increased 10 and 50 times, respectively, for illustrative purposes. The relative displacements of the fastener and skin in the y-direction, as indicated in Figure 8, are $-1.1 \times 10^{-2}$ mm and $3.5 \times 10^{-3}$ mm, respectively. In the first case, the deformation of the structure causes the countersunk bore to move upwards relative to the head of the bolt; the opposite occurs in the second case. The different magnitudes cause a significant difference in the strains in the coating for the two load cases. The assembly variables studied, and the preloads applied to the fasteners, further influenced the relative motion of the fastener head and skin, determining different levels of strain in the coating, albeit with less significant effects.

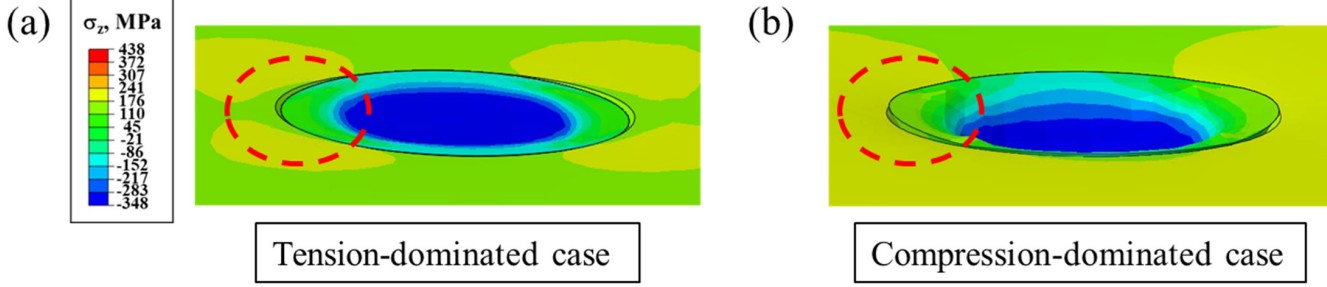

**Figure 8.** Exaggerated relative motion between the fastener head and plate surface for (**a**) the tension-dominated loading case and (**b**) the compression-dominated loading case.

The results from the finite element analysis (FEA) of the fastened plates and of the sub-modeling of the coating at the fastener–plate interface are presented in the following figures. The data are shown as the average strain at the primer and topcoat layers when different conditions are simulated: (a) a different preload of the fastener, Figure 9; (b) a different interference fit condition between the fastener and the skin hole, Figure 10; (c) a different loading condition, Figure 11.

The maximum strains occur along a circular contour in the primer and topcoat corresponding to the fastener head diameter. The data are extracted from the finite element model along a circular path that includes the highest strains. The resulting strains in the coating obtained around the fastener head for the case of a tension-dominated load in Figure 9 show a steep saddle profile for all preloads modeled. A 2% higher maximum strain was obtained on average for the case with the lowest preload. The maximum strains are near 10% for the primer, indicating that primer failure is expected to occur at the interface when tension-dominated loads are applied. The topcoat layer results are not affected significantly by the changes in the preload level. The principal strains on average are near 5% and failure might occur at the location of the highest strains. Plastic deformation is calculated by the model for the topcoat layer for all strain levels higher than 2.1% principal strain. This occurs because of the constraints on the topcoat layer and the value of the other two principal strains (not shown here). The ratio of maximum and minimum principal strains is −0.8 for the topcoat for the tension-dominated case.

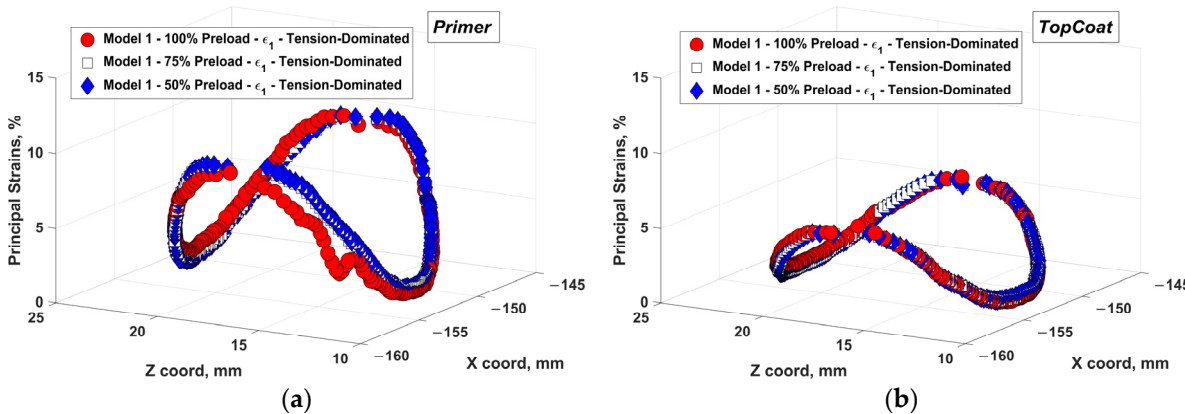

**Figure 9.** Resulting strain at the fastener–skin interface in the (**a**) primer and (**b**) topcoat layer of a coating system for tension-dominated loads and as a function of the fastener preload.

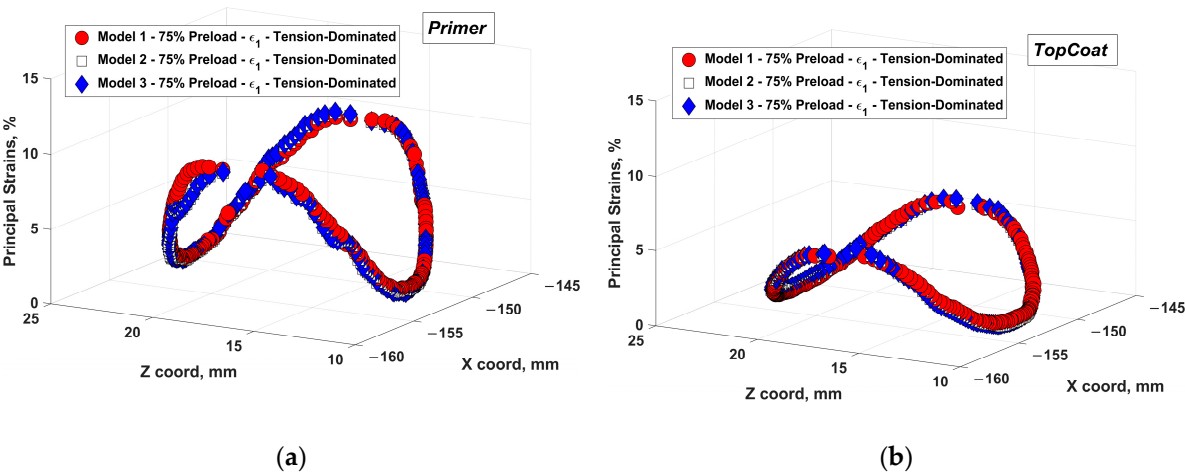

**Figure 10.** Resulting strain at the fastener–skin interface in the (**a**) primer and (**b**) topcoat layer of a coating system for severe loading conditions and for different levels of fastener interference fit, Models 1, 2, and 3, with 75% preload.

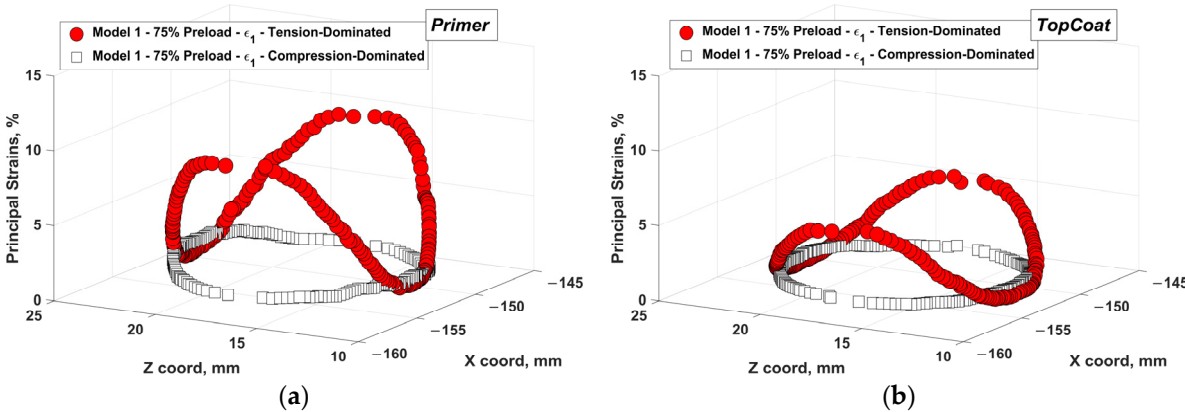

**Figure 11.** Resulting strain at the fastener–skin interface in the (**a**) primer and (**b**) topcoat layer of a coating system for tension-dominated and compression-dominated loads for Model 1, with 75% preload.

Figure 10 shows the results for Models 1, 2, and 3, all representing a preload condition of 75% and differing in their interference fit levels. The principal strain values for the primer and topcoat are very similar among the models. The primer results show principal strains 2% higher for the case of Model 3 (no interference fit); the topcoat results also show

5% higher principal strains for Model 3. This confirms the generally beneficial effect of interference fit fasteners.

Figure 11 compares the results of Model 1, with 75% preload, for two loading conditions: a tension-dominated and a compression-dominated load. The principal strains show very significant differences both in the maximum strains calculated and in the distribution of the strains along the circular path that corresponds to the head of the fastener. The strains for the compression-dominated load are around 2.6% for the primer and 1.2% for the topcoat. For both materials the relative differences between the minimum and maximum principal strains are much lower for the compression-dominated case than for the tension-dominated case: for the latter, the ratio of the highest to the smallest principal strain is 10.0 for the primer and 19.7 for the topcoat, and for the former, the ratio of the highest to lowest principal strains is 4.7 and 1.8, respectively, for the primer and topcoat.

*3.4. Coating Failure Models—Fracture in Unexposed Coatings*

In this section, the failure of the coating layer is analyzed as a function of the deformations calculated. The two load cases, representing tension- and compression-dominated conditions for the upper plate, are modeled for one condition of preload (75%) and interference fit ($15 \times 10^{-4} \frac{mm}{mm}$). The results for the tension-dominated loads are presented in Figure 12.

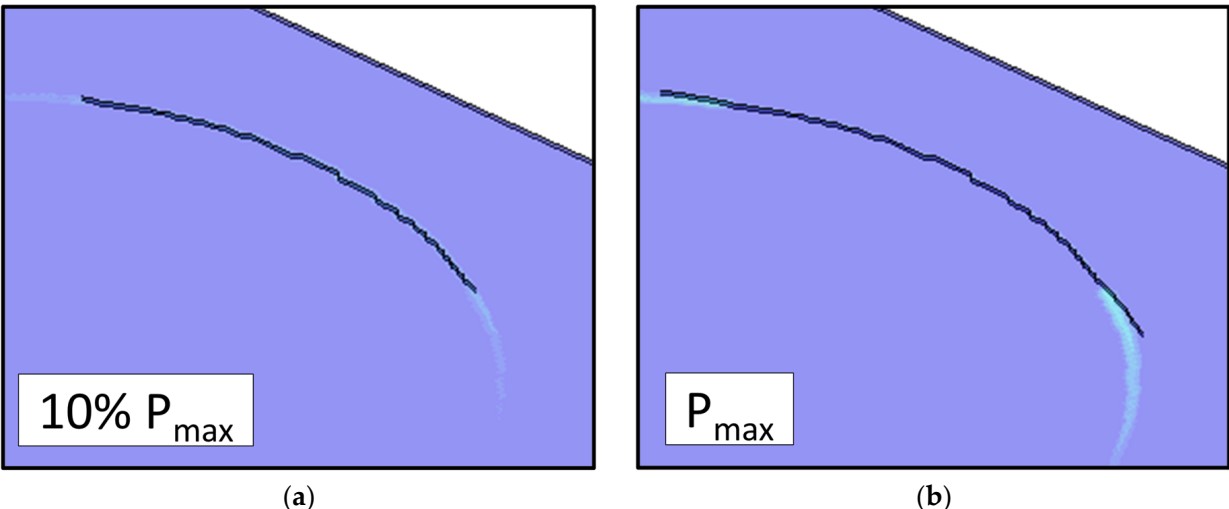

(**a**)         (**b**)

**Figure 12.** (**a**) Cracking of the primer layer at 10% of the maximum applied load and (**b**) crack propagation through the primer along the critical path of highest strain at the maximum applied load. Loads are obtained through simulations of tension-dominated events.

The principal strain contour plots for the primer layer are shown for 10% of the maximum applied tension load and for the maximum applied load, $P_{max}$. The crack initiates at the point of highest strain and propagates following the contour of strain concentration identified in Figure 9. Initial fracture occurs at the primer interface with the metal skin and progresses towards the outside layer of the coating system. The interface crack breaches the primer layer at between 10% and 15% of the maximum load, with the crack continuing to propagate along the perimeter of the fastener. In Figure 12a,b, the crack front crosses both the bottom and top face of the primer, and this is visualized using two distinct crack lines in the figure.

Figure 13 shows the beginning of the fracture of the topcoat layer, occurring at 55% of $P_{max}$, with crack initiation occurring only at the highest strain location and along the interface with the primer only. The crack front is visualized in the figure as a single crack line, showing that it has not breached the entire topcoat layer. The topcoat layer is not fully breached until the load reaches 95% of $P_{max}$. The fracture of the topcoat is localized in only

a few areas around the perimeter of the fastener. In Figure 13, this can be visualized by the appearance of small sections of a second crack line on the top surface of the topcoat.

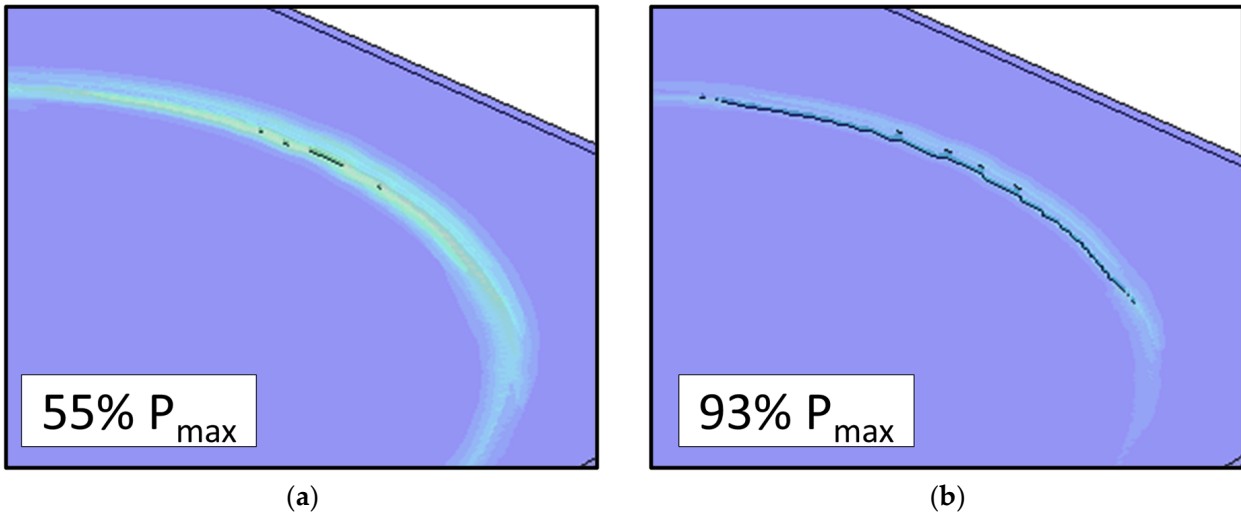

**(a)**                    **(b)**

**Figure 13.** (**a**) Cracking of the topcoat layer at 55% of the maximum applied load through propagation of the primer crack. (**b**) Propagation of the crack along the critical path of highest strain with only minimal breaching of the topcoat layer observed near the maximum applied load. Loads are obtained through simulations of tension-dominated events.

The results for the compression-dominated case are shown in Figure 14. Cracking does not occur in the primer until the applied loads reach 50% of the maximum. The applied strains are lower for the compression-dominated condition, but they are enough to cause fracturing of the primer layer. For the tension-dominated case, the ratio of maximum and minimum principal strains in the primer is −0.76, and for the compression-dominated case the ratio is −1. The latter is closer to a pure shear loading mode causing a reduction in the effective properties and a later occurrence of fracturing based on a maximum principal stress criterion [16]. For the compression-dominated case, the strain distribution is more uniform around the fastener and not specifically concentrated at two opposite edges of the fastener, as shown Figure 11. The crack in the primer follows the entire contour of the fastener head.

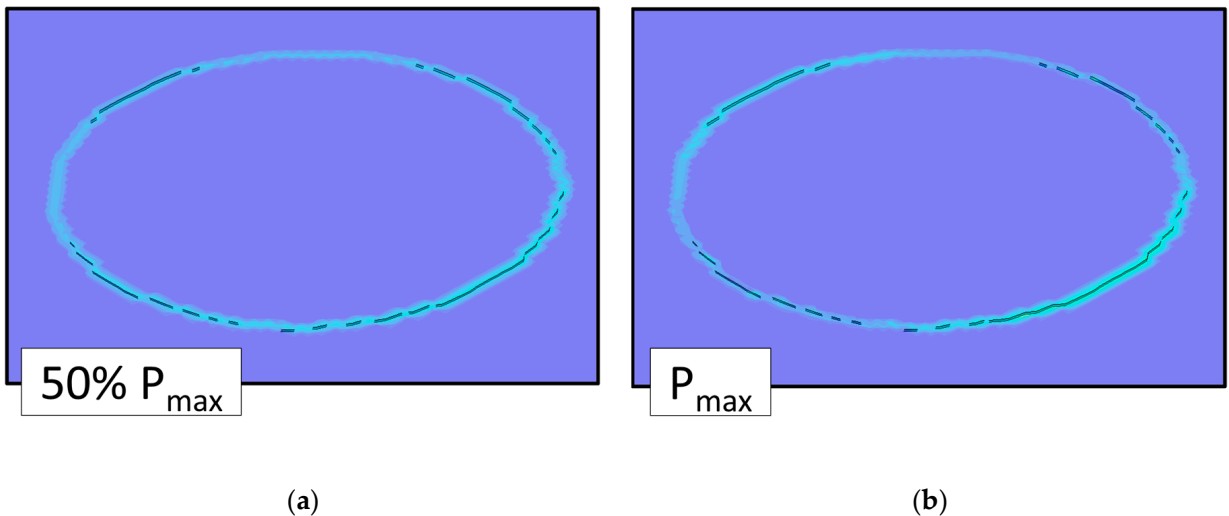

**(a)**                    **(b)**

**Figure 14.** (**a**) Cracking of the primer layer at 50% of the maximum applied load and (**b**) crack propagation through the primer along the critical path of highest strain at the maximum applied load. Loads are obtained through simulations of compression of the upper plate in Figure 4.

The topcoat layer is not breached by the crack in the primer as the loads are not significant enough to deform the topcoat layer and initiate a crack. Additionally, the energy released by the cracking occurring in the primer reduces the applied strains in the topcoat to below a critical level. Figure 15 shows the contour of the strain in the topcoat, with the maximum of the scale reaching 1%. Most of the strains in the topcoat are reduced well below 1%, with maximum values only reaching approximately 0.7% strain. Because of the reduced strains in the topcoat, the fracture does not propagate through the topcoat layer.

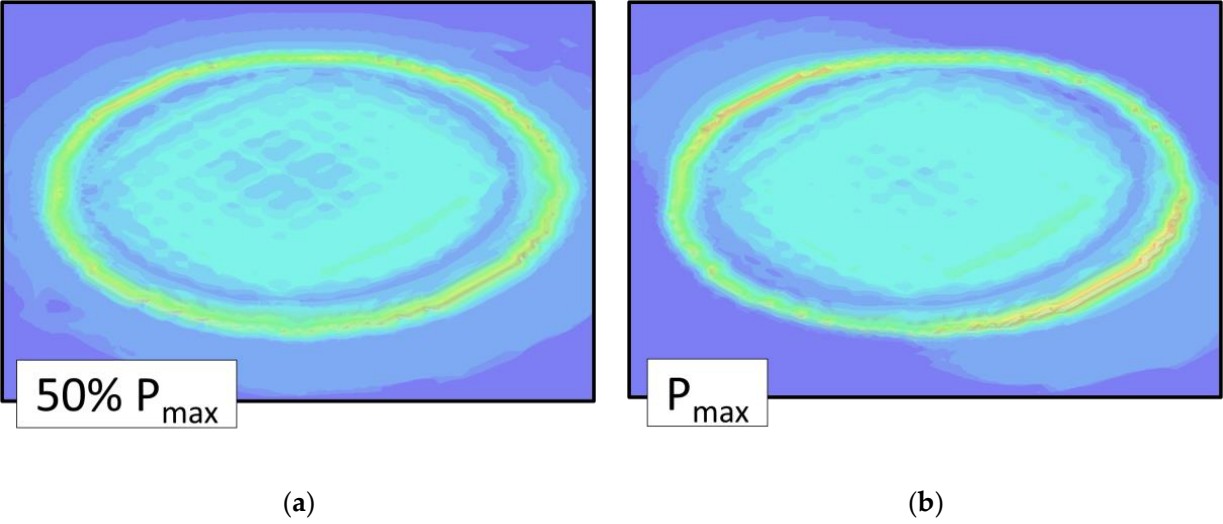

(**a**)                                                                      (**b**)

**Figure 15.** (**a**) Straining of the topcoat layer at 50% of the maximum applied load. (**b**) Strain of the topcoat layer observed at the maximum applied load. Loads are obtained from the compression-dominated case.

*3.5. Coating Failure Models—Fracturing in Aged Coatings*

In this section, we investigate loads that cause cracking of the topcoat layer using the model with the properties of the topcoat layer after long-term exposure to the atmospheric environment. Given the measured properties, the failure of the topcoat layer is expected to occur at a lower ratio of the maximum load during the loading cycles presented in Figure 13. Considering that the testing of the full system was not conclusive enough to support an estimation of the primer properties after long-term exposure and considering the elastic modulus of the full stack was generally consistent with the unexposed measured values, the primer properties used previously were not changed in the model. The loads from a tension-dominated event are used as the input to the model, and the cracking of the topcoat layer is monitored to calculate the load when cracking is initially observed in the topcoat layer and the loads when the topcoat layer is fully cracked.

Figure 16 shows that the initial occurrence of cracking in the topcoat layer occurs at approximately 38% of the maximum load caused by a tension-dominated event. A large extent of cracking comparable to the one obtained near the maximum load when the unexposed coating properties were used occurs at 67% of the maximum load. Both results, when compared to Figure 13, indicate the expected early cracking due to the degradation of properties in the topcoat layer measured experimentally.

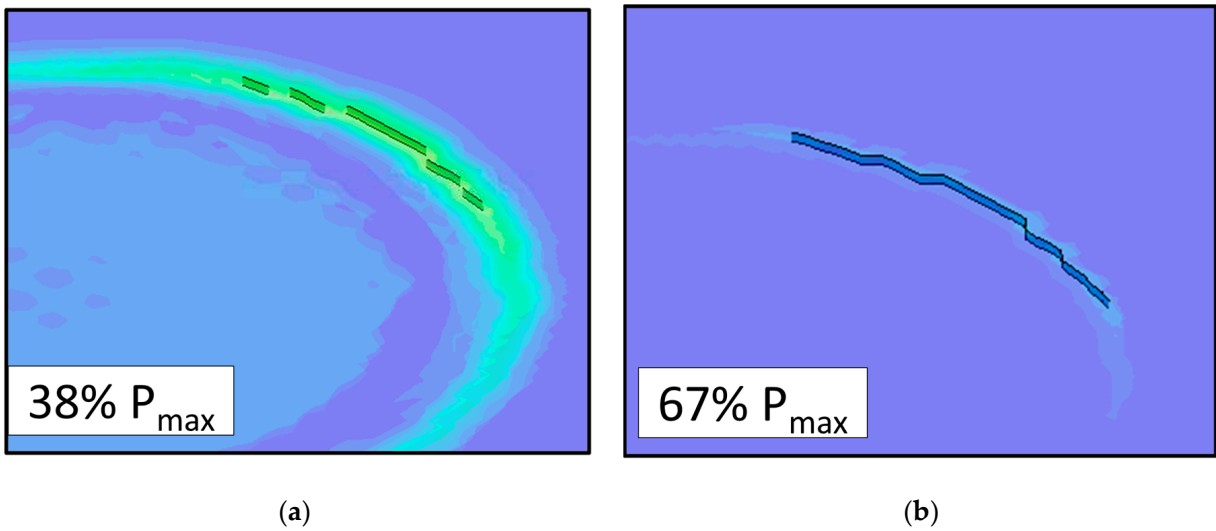

(**a**)                                                                    (**b**)

**Figure 16.** (**a**) Initial cracking of the topcoat layer at 38% of the maximum applied load by propagation of the primer crack and full breaching of the topcoat layer. (**b**) Propagation of the crack along the critical path of highest strain of the topcoat layer observed near 67% of the maximum applied load. Loads are obtained through simulations of tension-dominated events, and the cracking of the coating is symmetric around the fastener hole.

Figure 17 shows the contour of the strain in the topcoat for the compression-dominated load case, with the maximum of the scale reaching 1% for the material properties of the topcoat layer after long-term exposure to the atmospheric environment. The results are similar to the results for unexposed coating properties shown in Figure 15. Thus, this suggests that the local deformations are not sufficient to cause breaching of the topcoat even with degraded properties after long-term exposure. The margin to reach the strain leading to failure is reduced by approximately a factor of 2.0 with respect to the unexposed condition.

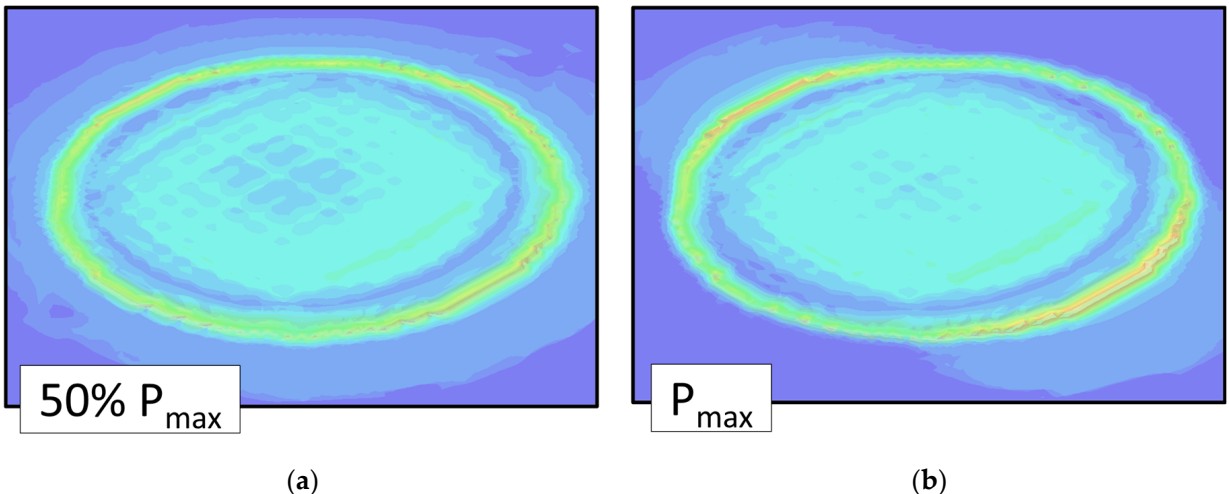

(**a**)                                                                    (**b**)

**Figure 17.** Results for the compression-dominated load case and topcoat layer properties after long-term exposure to the atmospheric environment. (**a**) Strain of the topcoat layer at 50% of the maximum applied load. (**b**) Strain of the topcoat layer observed under the maximum applied load.

*3.6. Coating Failure Models—Fatigue in Aged Coatings*

While the previous analysis indicates that a tension-dominated event might cause a direct failure of the coating system at the most critical interface, other locations which represent the majority of the coating areas are subjected to much lower strain levels. Figure 18a,b show the maximum principal strain in the primer and topcoat layers when subjected to

loads from a tension-dominated event. Figure 18c,d show the maximum principal strains for the compression-dominated case. Large areas of the primer are subjected to between 0.4% and 0.2% strain in the case of tension-dominated loads.

Thus, a second failure mode for the coating system is crack formation and propagation due to fatigue cycling. The strains are expected to cycle between the maximum values shown in Figure 18a–d and a minimum value corresponding to the mean static loads applied to the structure. However, these static loads are not estimated in this work.

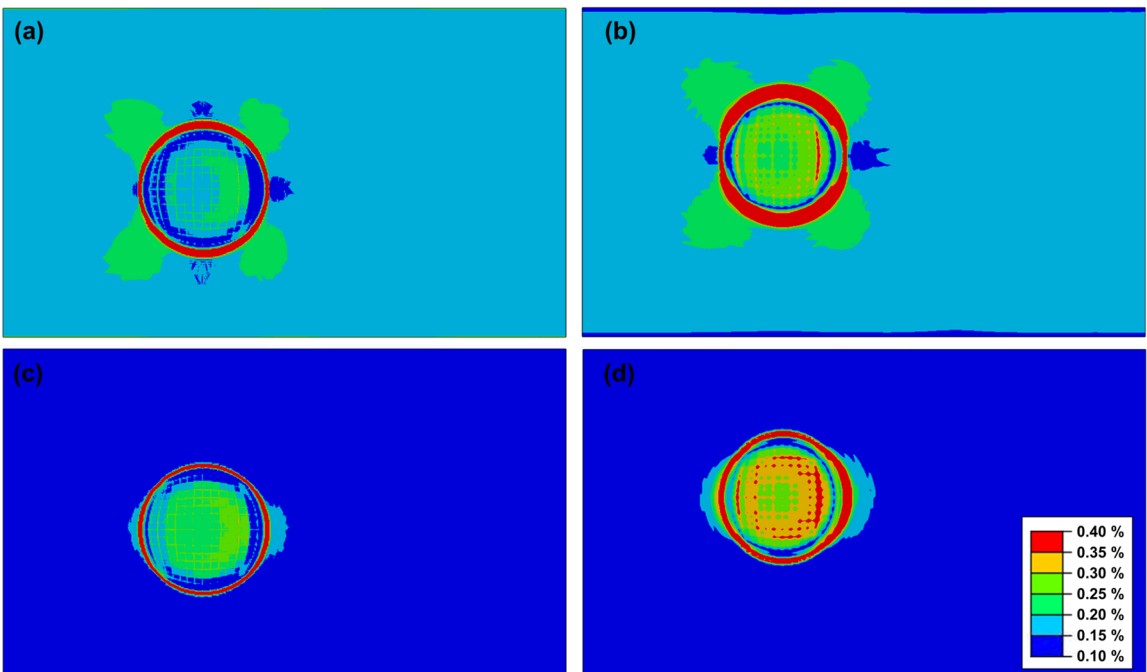

**Figure 18.** Maximum principal strain in the primer (**a**) and topcoat (**b**) layers of the coating systems with the application of loads from a tension-dominated event and for the case of compression-dominated loads (**c**,**d**). Strain levels are limited to 0.4% maximum and 0.1% minimum.

## 4. Discussion

As part of the qualification testing for coating systems, bend tests are used to ensure good adhesion to the substrate [17]. The resulting strains applied during these tests are usually high, which suggests that mechanical cracking of the full coating system only occurs for severe loading conditions. Thus, the tension- and compression-dominated load cases that were studied were proposed as representative cases during the service life of a joint located in a high-stress region of an aircraft, such as at a wing root. Based on the tensile properties collected and the fracture analysis performed, we can better qualify the extent of fracturing in the coating and the likelihood of cracking during the life of a fastened joint exposed to outdoor environmental conditions.

In our analysis, we found that cracking of the topcoat occurred only at or above 55% of the maximum applied tension-dominated loads. This suggests that complete fracturing of the coating layer likely only occurs for the most critical load cases at the most susceptible locations. However, while this is true for newly applied coatings, atmospheric exposure testing of the full coating system and topcoat materials showed that the topcoat layer was significantly affected by long-term environmental exposure. The results show both a reduction in elastic modulus and strain at failure. This reduction in elastic modulus may be due to moisture uptake and increased plasticization [18,19] and the reduction in failure strain from exposure to UV light. Results from the model analysis indicate that this degradation of the topcoat material properties caused the occurrence of initial and full coating breaching for a lower applied load ratio of 38% and 67% of the maximum loads studied, respectively.

In contrast, the fracture model results indicated that deformations from both tension- and compression-dominated loads were significant enough to cause fracturing of the primer layer. We observed that fracturing of the primer layer occurred at 10% and 50% of the respective maximum loads in the tension and compression-dominated load cases analyzed. Thus, despite using conservative applied loads, it seems very likely cracking would occur in the primer early in the life of the airframe, including at locations outside of the highest-stress regions.

Lastly, we also noted in our analysis that cracking at the interface between the primer and the metal skin caused a reduction in applied strain in the topcoat due to the released energy from the fracture. This implies that an already fractured primer might not transfer enough energy to cause fracturing of the topcoat at the same level observed in this work during a tension-dominated event. For example, as shown in Figure 14, a high-load event could establish a crack extending along the perimeter of the fastener head. This different initial condition is expected to influence cracking in the topcoat layer by reducing the available energy for fracture.

## 5. Conclusions

Free films of primer and topcoat organic polymers were tested in tension to obtain the material properties needed for a mechanical model of a fastener and skin interface coated with a coating system composed of the two polymers. The samples consisted of freshly cured free films and free films that underwent atmospheric exposure testing for 5736 h at an exposure test site in Fleming Key, FL. We used both sets of mechanical properties to model fracturing in a coating system applied to a two-plate aluminum structure with 20 stainless-steel fasteners. Analysis of the model calculations showed that fracturing of the primer is expected to occur upon application of low ratios of the maximum tension- and compression-dominated loads.

In contrast, for the freshly cured films, fracturing of the topcoat was only expected with the application of nearly the maximum tension load. However, exposure of the coatings to the atmospheric environment caused a deterioration in the topcoat mechanical properties. There was a 14.7% reduction in the modulus and a 46.3% reduction in the strain at fracture. The analysis with degraded properties identified an earlier occurrence of cracking for the topcoat when subjected to tension-dominated loads with respect to the case with unexposed properties, but no difference for the case of compression-dominated loads.

Thus, these results and analyses support the approach we have outlined in this manuscript. Coupling experimental measurements of mechanical properties of new and aged coatings with a finite element model of the fastener–skin–coating interface in highly stressed regions of an airframe will allow aircraft maintainers to make better estimates of the state of the coating system in response to flight operations, and ultimately to make better decisions on paint removal and repainting operations.

**Author Contributions:** Conceptualization, A.A., S.A.P., C.M.H., R.M.A. and E.B.I.; methodology, A.A.; software, A.A.; validation, S.A.P., R.M.A. and C.M.H.; investigation, A.A.; resources, E.B.I.; writing—original draft preparation, A.A.; writing—review and editing, A.A., R.M.A. and S.A.P.; visualization, A.A.; project administration, S.A.P.; funding acquisition, S.A.P., C.M.H., R.M.A., A.A. and E.B.I. All authors have read and agreed to the published version of the manuscript.

**Funding:** This work was sponsored by the Strategic Environmental Research and Development Program, SERDP, under project number WP19-1017.

**Institutional Review Board Statement:** Not applicable.

**Informed Consent Statement:** Not applicable.

**Data Availability Statement:** The original contributions presented in the study are included in the article, further inquiries can be directed to the corresponding author.

**Acknowledgments:** The views and conclusions contained herein are those of the authors and should not be interpreted as necessarily representing the official policies or endorsements, either expressed or implied, of the U.S. Naval Research Laboratory, SERDP, the U.S. Navy, or the U.S. government.

**Conflicts of Interest:** The authors declare no conflict of interest. The funders had no role in the design of the study; in the collection, analyses, or interpretation of data; in the writing of the manuscript; or in the decision to publish the results.

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
