# Peer review of "Tensile Properties of Aircraft Coating Systems and Applied Strain Modeling"

_coatings, doi:10.3390/coatings14010091_

Round 1

Reviewer 1 Report

Comments and Suggestions for Authors

The work is devoted to the study of the measured tensile properties of materials forming a coating system for aircraft structures before and after atmospheric exposure. In general, the presented work is quite interesting and promising not only from a fundamental point of view, but also from a practical one, since the results obtained in the future will make it possible to answer a number of questions related to the processes of corrosion and degradation, as well as the strength characteristics of materials. The presented work fully corresponds to the topic of the stated journal and can be accepted for publication after the authors answer the reviewer’s questions that arose while reading this article.

1. The choice of research objects is not entirely obvious. The authors should provide several compelling arguments about the reasons for choosing these structures, as well as their future prospects for application.

2. To complete the picture, the authors should expand the results of the work by comparing their results with other data, or with the results of corrosion tests for exposure to aggressive environments.

3. When describing mechanical tests, it is necessary to provide not only tensile curves, but also data on crack resistance and strength of coatings.

4. Authors should consider the possibility of presenting data on the properties of coatings, in particular morphological features and their changes caused by exposure to the atmosphere.

5. When describing the methodology for conducting experiments on aging or exposure to the atmosphere, the authors should explain the reasons for choosing the test time, what it was based on and whether there were prerequisites for conducting such studies.

Author Response

Dear Reviewer, please see our reply to your comments and questions in the attachment. Kind Regards

Reviewer 2 Report

Comments and Suggestions for Authors

This research work focused on studying experimentally the tensile properties of the materials forming a coating system for aircraft structures before and after atmospheric exposure. The coating system is composed of a MIL-PRF-85582E, Type II, Class C1, 2-part epoxy primer, and MIL-PRF-85285 Rev E, Type IV, Class H, 2-part polyurethane topcoat. The materials are deposited on a release paper and tested in tension. Samples are tested in the as-deposited condition after curing and after 5736 h of exposure to the atmospheric environment. The material properties for both unexposed and exposed conditions are used in a structural model to identify the applied loads that can lead to the cracking of the coating system at a critical fastener-skin interface location. After a revision of the paper, the decision is to accept this manuscript in its present form with a minor revision.

1.     There are some grammatical errors throughout the text. Minor editing of the English language is required.

2.     The author should pay attention to this phrase (Lines: 108/148/150/184/ 264/273/274/282/283/289/295/306/322/323/324/325/330/345/354/372/379/382/383/385/390/391/399/400/401/409/410/414/427/435/439/447/451/462/463/464/465/474/509/510) (Error! Reference source not found), and add adequate and appropriate references.

3.     The authors should mention each Figure and each table in the appropriate part of the text.

4.     The author mentioned in Figure 1. Atmospheric exposure of primer, topcoat, and full coating system samples at Fleming Key, 118 FL. In the case, of accelerated artificial aging tests, it is very important to study the simulated environmental conditions. Could the author detail and mention the environmental conditions that were studied?

5.     I invite the author to complete this work and enhance it in the future by studying the reaction mechanism between the materials forming a coating system and corrosive elements in the air stress-strain condition.

Comments on the Quality of English Language

There are some grammatical errors throughout the text. Minor editing of the English language is required.

Author Response

(The authors gave the same response as above.)

Reviewer 3 Report

Comments and Suggestions for Authors

Abstract must be rewriten in order to understand the aim of the paper, the obtained results and the used methods.

Complete the introduction with some aspects of the literature on how to joint aluminum-aluminum with stainless steel rivets to understand the materials used and explained in paragraph 2.4. Maybe this aspect ‘Note that aluminum alloy surfaces often receive a pretreatment or conversion coating that anodized the aluminum oxide layer to thicken and to aid in the adhesion between the surface and the primer.’ shall be developed in introduction

Line 20-22, please provide a bibliography or specify the material you are referring to. "Aluminum alloys sheets" or "fastened using steel fasteners to other aluminum alloy structures" is not enough.

Line 22 specify in technical terms "Countersunk fasteners are used on surfaces exposed to airflow, e.g., fastening wing skins to the underlying structure". Specify for the reader what those "Countersunk fasteners" are, etc

Lines 36-43 require references for understanding and argumentation

Line 63 specify a time interval/operations

Lines  75-78 specify the model used and the characteristics of the coating layers studied, and these should be found in the abstract so that an informed reader can go through the material clearly.

Lines 96-97, the deposition of coating layers is done according to ISO/ASTM, maybe it would be good to specify

Line 108, pay attention to the references, they are missing

Figure 1 is not edifying. I would suggest a drawing in which you can outline the deposited layers and the dimensions / type of deposition for UV testing. Illustrations of the brittle fractures explained in the text can also be presented.

Line 124 Specify the method of measuring of applied loads and displacements

Line 129 specify the grip method "in the grips" so that it does not break at the time of clamping

Line 136 4% and 2.5% as a percentage of how many pieces? Specify the number of parts subjected to tests.

Line 138, Chapter 2.4 Coating System Stress Field Model should be rewritten in accordance with the performed modeling. Von Mises stress distribution on the #6 rivet sample can be analyzed; then the entire model under discussion can be analyzed; FEM for simulating must be explained; explain deformed panel assembly

Some explanations from paragraph 2.4 should be transferred to the introduction.

Line  150 to redo the reference

Line 180 I suggest removing "small geometrical details" because the indicated area is the most exposed to stress.

Chapter 3.2 "tensile curves" comments refer to other articles.

Comment and argue “The strains at failure were consistently lower than previously measured for the full coating system.”

Line 493 ‘In our analysis, cracking of the topcoat was only found at, or above, 55% of the maximum applied tension-dominated loads.’ Reformulate

Conclusions must be rewritten

Line 544 – reference 1 must be upon journal indications

References are insignificant for the subject of the paper.

Please pay attention to the cross references/links in text

Author Response

(The authors gave the same response as above.)

Reviewer 4 Report

Comments and Suggestions for Authors

In this article, the authors provide and discuss experimentally obtained and modelled parameters, characterising tensile properties of the coating system, as well as individual components of this system (primer and topcoat), which are used for corrosion protection and other functions performance of aircraft structures. Undoubtedly, the data obtained after long-term exposure of the coating system in the natural atmosphere are particularly valuable. The manuscript is written in a very clear, coherent and well-argued way, justifying all the experiments and quite thoroughly discussing the obtained results of experimental studies and simulations. The work includes very practical aspects, which were worth to be investigated.

Although there are minor flaws in the manuscript, I believe that the material is suitable for publication in the section “Corrosion, Wear and Erosion” of the journal “Coatings” after minor revisions. My comments and suggestions that would improve the quality of the manuscript are as follows:

1) References to the tables and figures are omitted/not indicated properly in the text of the manuscript. Instead, it shows "Error! Reference source not found". Please provide, the necessary referencing information in the text of the manuscript;

2) Some statistical parameters (standard deviation and/or confidential interval) should be indicated next to the values of Elastic Modulus, Fracture Strain, Poisson's Ratio, Stress, and Plastic Strain (in Tables 1, 2, and 3), because as can be seen from the graphs in Figures 5 and 6, the dependence of Stress values on Strain for the samples of the same groups (Primer+TopCoat and Topcoat, unexposed and exposed to the atmospheric environment) show slightly different trends;

3) The manuscript should contain certain considerations, explanations or other information about the effect of the adhesion between the primer and top layer on the mechanical durability of the coating system Primer+TopCoat.

Author Response

(The authors gave the same response as above.)

Round 2

Reviewer 1 Report

Comments and Suggestions for Authors

The authors answered all the questions posed, the article can be accepted for publication.